# Inferring broken detailed balance in the absence of observable currents

Ignacio A. Martínez [1,5], Gili Bisker [2,5], Jordan M. Horowitz [3,4] & Juan M.R. Parrondo[1]

Identifying dissipation is essential for understanding the physical mechanisms underlying nonequilibrium processes. In living systems, for example, the dissipation is directly related to the hydrolysis of fuel molecules such as adenosine triphosphate (ATP). Nevertheless, detecting broken time-reversal symmetry, which is the hallmark of dissipative processes, remains a challenge in the absence of observable directed motion, flows, or fluxes. Furthermore, quantifying the entropy production in a complex system requires detailed information about its dynamics and internal degrees of freedom. Here we introduce a novel approach to detect time irreversibility and estimate the entropy production from time-series measurements, even in the absence of observable currents. We apply our technique to two different physical systems, namely, a partially hidden network and a molecular motor. Our method does not require complete information about the system dynamics and thus provides a new tool for studying nonequilibrium phenomena.

[1] Departamento de Estructura de la Materia, Física Termica y Electronica and GISC, Universidad Complutense de Madrid, 28040 Madrid, Spain. [2] Department of Biomedical Engineering, Faculty of Engineering, Center for Physics and Chemistry of Living Systems, Center for Nanoscience and Nanotechnology, Center for Light-Matter Interaction, Tel Aviv University, Tel Aviv 6997801, Israel. [3] Department of Biophysics, University of Michigan, Ann Arbor, MI 48109, USA. [4] Center for the Study of Complex Systems, University of Michigan, Ann Arbor, MI 48104, USA. [5] These authors contributed equally: Ignacio A. Martínez, Gili Bisker. Correspondence and requests for materials should be addressed to I.A.M. (email: iamartinez@ucm.es) or to G.B. (email: bisker@tauex.tau.ac.il)

Irreversibility is the telltale sign of nonequilibrium dissipation[1,2]. Systems operating far-from-equilibrium utilize part of their free energy budget to perform work, while the rest is dissipated into the environment. Estimating the amount of free energy lost to dissipation is mandatory for a complete energetics characterization of such physical systems. For example, it is essential for understanding the underlying mechanism and efficiency of natural Brownian engines, such as RNA-polymerases or kinesin molecular motors, or for optimizing the performance of artificial devices[3–5]. Often the manifestation of irreversibility is quite dramatic, signaled by directed flow or movement, as in transport through mesoscopic devices[6], traveling waves in nonlinear chemical reactions[7], directed motion of molecular motors along biopolymers[8], and the periodic beating of a cell's flagellum[9,10] or cilia[11]. This observation has led to a handful of experimentally validated methods to identify irreversible behavior by confirming the existence of such flows or fluxes[3,12–14]. However, in the absence of directed motion, it can be challenging to determine if an observed system is out of equilibrium, especially in small noisy systems where fluctuations could mask any obvious irreversibility[15]. One such possibility is to observe a violation of the fluctuation–dissipation theorem[16–18]; though this approach requires not just passive observations of a correlation function, but active perturbations in order to measure response properties, which can be challenging in practice. Thus, the development of noninvasive methods to quantitatively measure irreversibility and dissipation are necessary to characterize nonequilibrium phenomena.

Our understanding of the connection between irreversibility and dissipation has deepened in recent years with the formulation of stochastic thermodynamics, which has been verified in numerous experiments on meso-scale systems[19–22]. Within this framework, it is possible to evaluate quantities as the entropy along single nonequilibrium trajectories[23]. A cornerstone of this approach is the establishment of a quantitative identification of dissipation, or more specifically entropy production rate $\dot{S}$, as the Kullback–Leibler divergence (KLD) between the probability $\mathcal{P}(\gamma_t)$ to observe a trajectory $\gamma_t$ of length $t$ and the probability $\mathcal{P}(\tilde{\gamma}_t)$ to observe the time-reversed trajectory $\tilde{\gamma}_t$[1,24–29]:

$$\dot{S} \geq \dot{S}_{\mathrm{KLD}} \equiv \lim_{t \to \infty} \frac{k_{\mathrm{B}}}{t} D[\mathcal{P}(\gamma_t) || \mathcal{P}(\tilde{\gamma}_t)], \qquad (1)$$

where $k_{\mathrm{B}}$ is Boltzmann's constant. The KLD between two probability distributions $p$ and $q$ is defined as $D[p||q] \equiv \sum_x p(x) \ln (p(x)/q(x))$ and is an information-theoretic measure of distinguishability[30]. For the rest of the paper we take $k_{\mathrm{B}} = 1$, so the entropy production rate has units of time$^{-1}$. The entropy production $\dot{S}$ in Eq. (1) has a clear physical meaning. It is the usual entropy production defined in irreversible thermodynamics by assuming that the reservoirs surrounding the system are in equilibrium. For instance, in the case of isothermal molecular motors hydrolyzing ATP to ADP+P at temperature $T$, the entropy production in Eq. (1) is $\dot{S} = r\Delta\mu/T - \dot{W}/T$, where $r$ is the ATP consumption rate, $\Delta\mu = \mu_{\mathrm{ATP}} - \mu_{\mathrm{ADP}} - \mu_{\mathrm{P}}$ is the difference between the ATP, and the ADP and P chemical potentials, and $\dot{W}$ is the power of the motor[31]. In many experiments, all these quantities can be measured except the rate $r$. Therefore, the techniques that we develop in this paper can help to estimate the ATP consumption rate, even at stalling conditions.

The equality in Eq. (1) is reached if the trajectory $\gamma_t$ contains all the meso- and microscopic variables out of equilibrium. Hence the relative entropy in Eq. (1) links the statistical time-reversal symmetry breaking in the mesoscopic dynamics directly to dissipation. Based on this connection, estimators of the relative entropy between stationary trajectories and their time reverses allow one to determine if a system is out of equilibrium or even bound the amount of energy dissipated to maintain a

nonequilibrium state. Such an approach, however, is challenging to implement accurately as it requires large amounts of data, especially when there is no observable current[32].

Despite the absence of observable average currents, irreversibility can still leave a mark in fluctuations. Consider, for example, a particle hoping on a 1D lattice, as in Fig. 1, where up and down jumps have equal probabilities, but the timing of the jumps have different likelihoods. Although there is no net drift on average, the process is irreversible, since any trajectory can be distinguished from its time reverse due to the asymmetry in jump times. Thus, beyond the sequence of events, the timing of events can reveal statistical irreversibility. Such a concept was used, for example, to determine that the *E. Coli* flagellar motor operates out of equilibrium based on the motor dwell-time statistics[33].

In this work, we establish a technique that allows one to identify and quantify irreversibility in fluctuations in the timing of events, by applying Eq. (1) to stochastic jump processes with arbitrary waiting time distributions, that is, semi-Markov processes, also known as continuous time random walks (CTRW) in the context of anomalous diffusion. Such models emerge in a plethora of contexts[34–36] ranging from economy and finance[37] to biology, as in the case of kinesin dynamics[38] or in the anomalous diffusion of the Kv2.1 potassium channel[39]. In fact, as we show below and in the Methods section, semi-Markov processes result in experimentally relevant scenarios where one has access only to a limited set of observables of Markov kinetic networks with certain topologies. We begin by reviewing the semi-Markov framework, where we present our main result of the entropy production rate estimator. Next, we apply our approach to general hidden networks, where an observer has access only to a subset of the states, comparing our estimator with previous proposals for partial entropy production that are zero in the absence of currents. Finally, we address a particularly important case of molecular motors, where their translational motion is easily observed, but the biochemical reactions that power their motion are hidden. Remarkably, our technique allows us to even reveal the existence of parasitic mechano-chemical cycles at stalling—where the observed current vanishes or the motor is stationary—simply from the distribution of step times. In addition, our quantitative lower bound on the entropy production rate can be used to shed light on the efficiency of molecular motors operation and on the entropic cost of maintaining their far-from-equilibrium dynamics[40–44].

## Results

**Irreversibility in semi-Markov processes**. A semi-Markov process is a stochastic renewal process $\alpha(t)$ that takes values in a discrete set of states, $\alpha = 1, 2, \ldots$. The renewal property implies

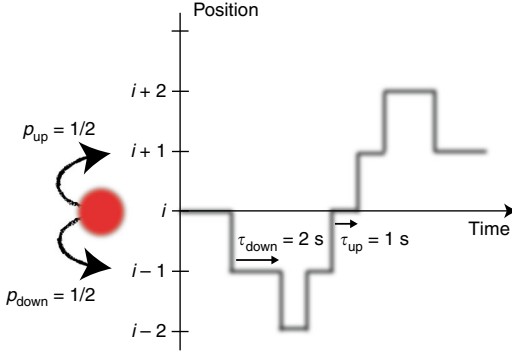

**Fig. 1** Brownian particle jumping on an one-dimensional lattice. Jumps up and down are equally likely, but with asymmetric jump rates. As a result, the irreversibility of the dynamics is contained solely in the timing fluctuations

that the waiting time intervals $t_\alpha$ in a given state $\alpha$ are positive, independent, and identically distributed random variables. If the system arrives to state $\alpha$ at $t = 0$, the probability to jump to a different state $\beta$ at time $[t, t + dt]$ is $\psi_{\beta\alpha}(t)dt$, with $\psi_{\beta\alpha}(t)$ being the probability density of transition times[45]. These densities are not normalized, with $p_{\beta\alpha} \equiv \int_0^\infty \psi_{\beta\alpha}(t)dt$ being the probability for the next jump to be $\alpha \to \beta$ given that the walker arrived at $\alpha$. We assume that the particle eventually leaves any site $\alpha$, i.e., $\psi_{\alpha\alpha}(t) = 0$ and $\sum_\beta p_{\beta\alpha} = 1$, so the matrix $p_{\beta\alpha}$ is a stochastic matrix. Its normalized (right) eigenvector $R_\alpha$ with eigenvalue 1, then represents the fraction of visits to each state $\alpha$.

The waiting time distribution at site $\alpha$, $\psi_\alpha(t) = \sum_\beta \psi_{\beta\alpha}(t)$, is normalized with average waiting time $\tau_\alpha$. We can also define the waiting time distribution conditioned on a given jump $\alpha \to \beta$ as $\psi(t|\alpha \to \beta) \equiv \psi_{\beta\alpha}(t)/p_{\beta\alpha}$, which is already normalized.

Consider now a generic semi-Markovian trajectory $\gamma_t$ of length $t$ with $n$ jumps, which is fully described by the sequence of jumps and jump times, $\gamma_t = \{\alpha_1 \xrightarrow{t_1} \alpha_2 \xrightarrow{t_2} \ldots \xrightarrow{t_{n-1}} \alpha_n \xrightarrow{t_n} \alpha_{n+1}\}$ with $\sum_n t_n = t$, occurring with probability $\mathcal{P}(\gamma_t) = \psi_{\alpha_2,\alpha_1}(t_1)\psi_{\alpha_3,\alpha_2}(t_2) \ldots \psi_{\alpha_{n+1},\alpha_n}(t_n)$. In order to characterize the dissipation of this single trajectory, we must define its time reverse $\tilde{\gamma}_t = \{\alpha_n \xrightarrow{t_n} \alpha_{n-1} \xrightarrow{t_{n-1}} \ldots \xrightarrow{t_2} \alpha_1 \xrightarrow{t_1} \alpha_0\}$ whose probability is given by $\mathcal{P}(\tilde{\gamma}_t) = \psi_{\alpha_0,\alpha_1}(t_1) \ldots \psi_{\alpha_{n-1},\alpha_n}(t_n)$, see Methods and Fig. 5.

Directly applying Eq. (1) to this scenario shows that the KLD between the probability distributions of the forward and backward trajectories can be split into two contributions (see Methods):

$$\dot{S}_{\text{KLD}} = \dot{S}_{\text{aff}} + \dot{S}_{\text{WTD}}. \qquad (2)$$

The first term, $\dot{S}_{\text{aff}}$, or affinity entropy production, results entirely from the divergence between the state trajectories, regardless of the jump times, $\sigma \equiv \{\alpha_1, \alpha_2, \ldots, \alpha_{n+1}\}$ and $\tilde{\sigma} \equiv \{\alpha_n, \ldots, \alpha_1, \alpha_0\}$, that is, it accounts for the affinity between states:

$$\dot{S}_{\text{aff}} = \frac{1}{\mathcal{T}} \sum_{\alpha\beta} p_{\beta\alpha} R_\alpha \ln \frac{p_{\beta\alpha}}{p_{\alpha\beta}} = \frac{1}{\mathcal{T}} \sum_{\alpha < \beta} J_{\beta\alpha}^{\text{ss}} \ln \frac{p_{\beta\alpha}}{p_{\alpha\beta}}, \qquad (3)$$

where $J_{\beta\alpha}^{\text{ss}} = p_{\beta\alpha} R_\alpha - p_{\alpha\beta} R_\beta$ is the net probability flow per step, or current, from $\alpha$ to $\beta$, and the factor $\mathcal{T} = \sum_\alpha \tau_\alpha R_\alpha$ is the mean duration of each step, which can be used to transform the units from per-step to per-time[46]. We see that the affinity entropy production vanishes in the absence of currents, as it occurs in arbitrary Markov systems[32,47].

The contribution due to the waiting times is expressed in terms of the KLD between the waiting time distributions

$$\dot{S}_{\text{WTD}} = \frac{1}{\mathcal{T}} \sum_{\alpha\beta\mu} p_{\mu\beta} p_{\beta\alpha} R_\alpha D[\psi(t|\beta \to \mu) || \psi(t|\beta \to \alpha)], \qquad (4)$$

which is the main result of this paper and allows one to detect irreversibility in stationary trajectories with zero current.

Notice that $R_\alpha$ being the occupancy of state $\alpha$, $p_{\beta\alpha} R_\alpha$ is the probability to observe the sequence $\alpha \to \beta$ in a stationary forward trajectory, while $p_{\mu\beta} p_{\beta\alpha} R_\alpha$ is the probability to observe the sequence $\alpha \to \beta \to \mu$.

Equation (2) is the chain rule of the relative entropy applied to the semi-Markov process and the core of our proposed estimator. In the special case of Poisson jumps, $D[\psi(t|\beta \to \mu) || \psi(t|\beta \to \alpha)] = 0$ since all waiting time distributions for jumps starting at a given site $\beta$ are equal (see Methods), and we recover the standard expression for the relative entropy of Markov processes $\dot{S} = \dot{S}_{\text{aff}}$. It is worth mentioning that previous attempts to establish the entropy production of semi-Markov processes failed to identify

the term $S_{\text{WTD}}$ because they assumed that the waiting time distributions were independent of the final state, as occurs in Markov processes[48–50]. However, such a strong assumption does not hold in many situations of interest, as in the ones discussed below.

**Decimation of Markov chains and second-order semi-Markov processes.** Semi-Markov processes appear when sites are decimated from Markov chains of certain topologies. Figure 2 shows representative examples. In Fig. 2a, b, we show two models of a molecular motor that runs along a track with sites $\{\ldots, i - 1, i, i + 1, \ldots\}$ and has six internal states. If the spatial jumps (red lines) and the transitions between internal states (black lines) are Poissonian jumps, then the motor is described by a Markov process. On the other hand, when the internal states are not accessible to the experimenter, the waiting time distributions corresponding to the spatial jumps $i \to i \pm 1$ are no longer exponential and the motion of the motor must be described by a semi-Markov process. Figure 2a shows an example where the decimation of internal states directly yields a semi-Markov process ruling the spatial motion of the motor. The second example, sketched in Fig. 2b, is more involved since the upward and the downward jumps end in different sets of internal states. As a consequence, the waiting time distribution of, say, the jump $i \to i + 1$, depends on the site that the motor visited before site $i$. Then, the resulting dynamics must be described by a second-order semi-Markov process, that is, one has to consider the states $\alpha(t) = [i_{\text{prev}}(t), i(t)]$, where $i(t)$ is the current position of the motor and $i_{\text{prev}}(t)$ is the previous position, right before the jump.

The same applies to generic kinetic networks, as the one depicted in Fig. 2c. Suppose that the original network is Markovian with states $i = 1, \ldots, 5$. However, if the experimenter only has access to states 1 and 2, with the rest clumped together into a hidden state $H$, then the resulting dynamics is also a second-order semi-Markov process with the reduced set $i = 1, 2, H$.

For second-order semi-Markov processes the affinity entropy production reads

$$\dot{S}_{\text{aff}} = \frac{1}{\mathcal{T}} \sum_{i,j,k} p(ijk) \ln \frac{p([ij] \to [jk])}{p([kj] \to [ji])}, \qquad (5)$$

where $p(ijk) \equiv R_{[ij]} p([ij] \to [jk])$ is the probability to observe the sequence $i \to j \to k$. This entropy is still proportional to the current

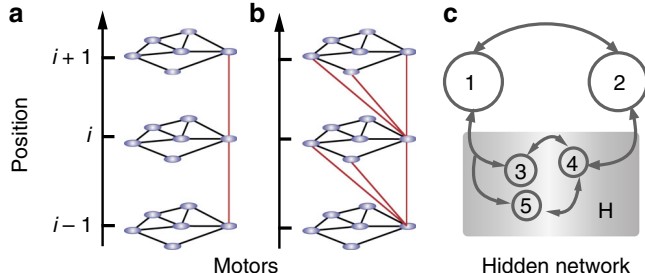

**Fig. 2** Decimation of Markov processes. **a**, **b** Molecular motor model: An observer with access only to the position (vertical axis) cannot resolve the internal states (circles). **a** Decimation to position results in a first-order Markov process, since spatial jumps connect the same internal state. **b** Decimation results in a second-order semi-Markov process, where the waiting time distribution for spatial transitions depends on whether the motor previously jumped down or up. **c** Hidden kinetic network: An observer unable to resolve states 3, 4, and 5, treats them as a single hidden state $H$. The resulting decimated network is a second-order semi-Markov process on the three states 1, 2, and $H$, where the non-Poissonian waiting time distributions for transitions out of state $H$ depend on the past

for one-dimensional processes and therefore vanishes in the absence of flows in the observed dynamics, see Methods. The entropy production contribution due to the irreversibility of the waiting time distributions is

$$\dot{S}_{\mathrm{WTD}} = \frac{1}{\mathcal{T}} \sum_{i,j,k} p(ijk) D[\psi(t|[ij] \rightarrow [jk]) || \psi(t|[kj] \rightarrow [ji])]. \quad (6)$$

Let us emphasize that the calculation of $\dot{S}_{\mathrm{WTD}}$ requires collecting statistics on sequences of two consecutive jumps, i.e., $i \rightarrow j \rightarrow k$. We now proceed to apply these results to generic cases of simple kinetic networks and molecular motors.

**Hidden networks**. We first apply our formalism to estimate the dissipation in kinetic networks with hidden states, which have received increasing attention in recent years owing to their many practical and experimental implications[24,32,47,51–53].

Consider a network where $\omega_{ij}$ is the transition rate from state $j$ to state $i$, with $\pi_i$ the steady-state distribution. The total entropy production rate at steady-state is[54]

$$\dot{S} = \sum_{i<j} \left( \omega_{ji}\pi_i - \omega_{ij}\pi_j \right) \ln \frac{\omega_{ji}\pi_i}{\omega_{ij}\pi_j}, \quad (7)$$

where the positivity of $\dot{S}$ stems from the positivity of each individual term in the sum[40,52,55]. In order to calculate the total entropy production $\dot{S}$ according to Eq. (7), full knowledge of the steady-state probability distribution $\{\pi_i\}$ and the transition rates between all the microstates $\{\omega_{ij}\}$ is required. We would like to assign a partial entropy production rate when one only has access to a limited set of states and transitions. To be concrete, we focus on the scenario depicted in Fig. 2c, where only states 1 and 2 can be observed. Previously, two approaches for assigning partial entropy production rate in such a case have been defined in the literature, both of which provide a lower bound on the total entropy production rate[56]: the passive partial entropy production rate due to Shiraishi and Sagawa[52], and the informed partial entropy production rate due to Polettini and Esposito[53,57]. The passive partial entropy production rate $\dot{S}_{\mathrm{PP}}$ for the single observed link is simply given by the corresponding term in Eq. (7)

$$\dot{S}_{\mathrm{PP}} = (\omega_{12}\pi_2 - \omega_{21}\pi_1) \ln \frac{\omega_{12}\pi_2}{\omega_{21}\pi_1}, \quad (8)$$

where the observer is assumed to have access to the steady-state populations of the two states, $\pi_1$ and $\pi_2$, as well as the transition rates between them.

The informed partial entropy production $\dot{S}_{\mathrm{IP}}$ for the single link requires additional information: the observer is assumed to have control over the transition rates of the observed link, without affecting any of the hidden transitions, such that they can stall the corresponding current and record the ratio of populations in the two observed states, $\pi_1^{\mathrm{stall}}/\pi_2^{\mathrm{stall}}$. The stalling distribution $\pi_i^{\mathrm{stall}}$ produces an effective thermodynamic description of the observed subsystem[53] and an effective affinity with which the informed partial entropy production rate is calculated:

$$\dot{S}_{\mathrm{IP}} = (\omega_{12}\pi_2 - \omega_{21}\pi_1) \ln \frac{\omega_{12}\pi_2^{\mathrm{stall}}}{\omega_{21}\pi_1^{\mathrm{stall}}}. \quad (9)$$

Although the informed partial entropy production was proven to produce a better estimation of the total dissipation compared with the passive partial entropy production, i.e., $\dot{S}_{\mathrm{PP}} \leq \dot{S}_{\mathrm{IP}} \leq \dot{S}$[56], both vanish at stalling conditions. Hence, even if the system is in a nonequilibrium steady-state, when the current over the observed link is zero, these estimators cannot give a nontrivial lower bound on the total entropy production. To be fair, we point out that each estimator uses different information.

For the KLD estimator, we assume that the observer can record whether the system is in states 1 or 2, or in the hidden part of the network, $H$, which is a coarse-grained state representing the unobserved subsystem. In this case, the resulting contracted network has three states, $\{1, 2, H\}$. Jumps between states 1 and 2 follow Poissonian statistics, as in a general continuous-time Markov process, with the same rates as in the original network. On the other hand, jumps from $H$ to 1 or 2 are not Poissonian and depend on the state just prior to entering the hidden part. To apply our results for semi-Markov processes, we thus have to consider the states $\alpha(t) = [i_{\mathrm{prev}} \ i(t)]$, where $i(t) = 1, 2, H$ is the current state and $i_{\mathrm{prev}}(t) = 1, 2, H$ is the state right before the last jump. To make the equations more compact, we will use the short-hand notation $_i j \equiv [i \ j]$ for the remainder of this section.

Similar to Eq. (2), the semi-Markov entropy production rate for hidden networks, $\dot{S}_{\mathrm{KLD}}$, consists of two contributions: the affinity estimator $\dot{S}_{\mathrm{aff}}$ and the WTD estimator $\dot{S}_{\mathrm{WTD}}$. In this case, the affinity estimator, Eq. (5), is given by

$$\dot{S}_{\mathrm{aff}} = \frac{J_{21}^{\mathrm{ss}}}{\mathcal{T}} \ln \frac{p(_1 2 \rightarrow _2 H)p(_2 H \rightarrow _H 1)p(_H 1 \rightarrow _1 2)}{p(_1 H \rightarrow _H 2)p(_H 2 \rightarrow _2 1)p(_2 1 \rightarrow _1 H)}, \quad (10)$$

where $J_{21}^{\mathrm{ss}}$ is the stationary current per step from 1 to 2, defined as $J_{21}^{\mathrm{ss}} = R_{[12]} - R_{[21]}$. As expected, this term vanishes when detailed balance holds and the current is zero (see Methods). Applying Eq. (6) to the semi-Markov process results in the following expression for the contribution of the hidden estimator

$$\dot{S}_{\mathrm{WTD}} = \frac{p(1H2)}{\mathcal{T}} D[\psi(t|_1 H \rightarrow _H 2) || \psi(t|_2 H \rightarrow _H 1)] \\ + \frac{p(2H1)}{\mathcal{T}} D[\psi(t|_2 H \rightarrow _H 1) || \psi(t|_1 H \rightarrow _H 2)], \quad (11)$$

where $p(ijk) = R_{[ij]}p(j \rightarrow _j k)$. In Methods, we further show that for a network of a single cycle of states the informed partial entropy production $\dot{S}_{\mathrm{IP}}$ equals the affinity estimator $\dot{S}_{\mathrm{aff}}$ defined in Eq. (5). Summarizing, we have the hierarchy $\dot{S}_{\mathrm{PP}} \leq \dot{S}_{\mathrm{IP}} = \dot{S}_{\mathrm{aff}} \leq \dot{S}_{\mathrm{KLD}} \leq \dot{S}$.

Let us apply the hidden semi-Markov entropy production framework to a specific example of a network with four states, two of which are hidden (Fig. 3a). We have chosen a random $4 \times 4$ matrix, with non-negative off diagonal entries and zero sum columns, as a generator of a continuous-time Markov jump process over the four states. The rates over the observed link were varied according to $\omega_{12}(F) = \omega_{12}e^{\beta FL}$ and $\omega_{21}(F) = \omega_{21}e^{-\beta FL}$ over a range of values of a force $F$ that included the stalling force $F^{\mathrm{stall}}$, where $\beta = 1/T$ is the inverse temperature and $L$ is a characteristic length scale. For each value of $F$, we contracted the dynamics to the three states, 1, 2, and $H$ (Fig. 3b, c), and estimated the waiting time distributions $\psi(t|_2 H \rightarrow _H 1)$ and $\psi(t|_1 H \rightarrow _H 2)$ using a kernel density estimate with a positive support[58,59] (see Methods), depicted in Fig. 3d. From those distributions, we derived the hidden semi-Markov entropy production rate $\dot{S}_{\mathrm{KLD}}$ (Fig. 3e). We further calculated both the passive- and informed-partial entropy production rates to compare all the estimators to the total entropy production rate (Fig. 3e). Our results clearly demonstrate the advantage of using the waiting time distributions for bounding the total entropy production rate compared with the two other previous approaches. Our framework can reveal the irreversibility and the underlying dissipation, even when the observed current vanishes, without the need of manipulating the system.

The KLD entropy production rate was also estimated from simulated experimental data, obtained by sampling random trajectories of $10^7$ jumps using the Gillespie algorithm[60]. The simulated trajectories (Fig. 3b) were coarse-grained into the set of states of the hidden semi-Markov model (Fig. 3c), and the hidden semi-Markov entropy production rate for the simulated experimental data, $\dot{S}_{\mathrm{KLD}}^{\mathrm{Exp}}$, was estimated as above (Fig. 3e, blue crosses).

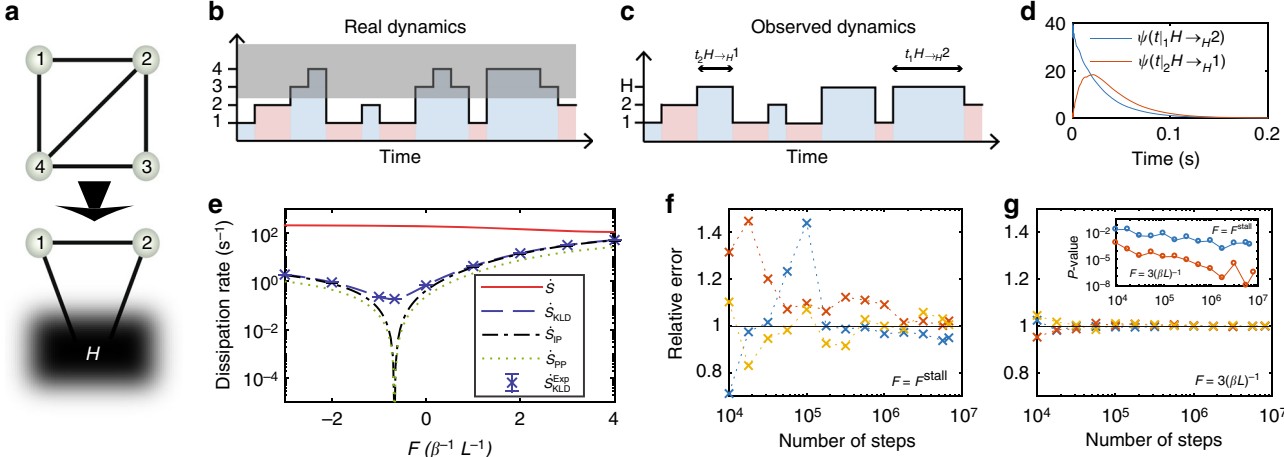

**Fig. 3 Hidden network. a** Four-state network, as seen by an observer, with access only to states 1, 2, $H$. **b**, **c** Illustration of a trajectory over the four possible states (**b**) where the gray region corresponds to the hidden part. The resulting observed semi-Markov dynamics (**c**). **d** Kernel density estimation of the wait time distributions at $F = F^{stall}$. **e** Estimated total entropy production rate $\dot{S}$ (solid red line), entropy production for semi-Markov model $\dot{S}_{KLD}$ (dashed blue curve), informed partial entropy production rate $\dot{S}_{IP}$ (dashed-dotted black curve), the passive partial entropy production rate $\dot{S}_{PP}$ (dotted green curve), and the experimental entropy production rate estimated according to the semi-Markov model $\dot{S}_{KLD}^{Exp}$ (blue crosses). **f**, **g** Relative error (ratio of experimental entropy production rate to analytical value) for three random trajectories as a function of the number of steps at $F = F^{stall}$ (**f**) and $F = 3\beta^{-1}L^{-1}$ (**g**) showing faster convergence away from the stalling force. Inset: $p$-value for rejecting the null hypothesis that the experimental data was sampled from a zero mean distribution as a function of the number of steps for $F = F^{stall}$ (blue curve), and $F = 3\beta^{-1}L^{-1}$ (red curve), showing that the average is statistically significant different from zero. The numerical simulations were done using the Gillespie algorithm with the following transition rates: $\omega_{12} = 2$ s$^{-1}$, $\omega_{13} = 0$ s$^{-1}$, $\omega_{14} = 1$ s$^{-1}$, $\omega_{21} = 3$ s$^{-1}$, $\omega_{23} = 2$ s$^{-1}$, $\omega_{24} = 35$ s$^{-1}$, $\omega_{31} = 0$ s$^{-1}$, $\omega_{32} = 50$ s$^{-1}$, $\omega_{34} = 0.7$ s$^{-1}$, $\omega_{41} = 8$ s$^{-1}$, $\omega_{42} = 0.2$ s$^{-1}$, $\omega_{43} = 75$ s$^{-1}$, where the diagonal elements were chosen to have zero sum coloums

In order to assess the rate of convergence with increasing number of simulated steps, we calculated the $\dot{S}_{KLD}^{Exp}$ for different fractions of the $10^7$ steps trajectories, showing <20% error above $10^5$ steps at stalling, and <5% error away from stalling for trajectories with as little as $10^4$ steps (Fig. 3f, g). Let us stress that the hidden semi-Markov entropy production rate averaged over three simulated experimental trajectories produced a lower bound on the total entropy production rate, which was strictly positive and statistically significant different from zero ($p < 0.05$, Fig. 3g, inset) for all trajectory lengths tested.

**Molecular motors**. A slight modification of the case analyzed in the previous section allows us to study molecular motors with hidden internal states. We are interested in the schemes previously sketched in Fig. 2a, b, where a motor can physically move in space or switch between internal states. The observed motor position is labeled by $\{..., i-1, i, i+1, ...\}$. All jumps are Poissonian and obey local detailed balance, with an external source of chemical work, $\Delta\mu$, and an additional mechanical force $F$ that can act only on the spatial transitions.

Analogous to the previous example, the observed dynamics is a second-order semi-Markov process. To make the following equations more intuitive, we use the graphical notation ⌐ for two consecutive upward jumps ($i-1 \rightarrow i \rightarrow i+1$), ⌐ for a downward jump followed by and upward one, ⌐ for an upward followed by a downward jump, and ⌐ for two consecutive downward jumps. Notice that the probabilities are normalized as $p_{⌐} + p_{⌐} = p_{⌐} + p_{⌐} = 1$.

Similar to Eq. (2), we have the decomposition of the KLD estimator into a contribution from state affinities given by

$$\dot{S}_{aff} = \frac{J^{ss}}{\mathcal{T}} \ln \frac{p_{⌐}}{p_{⌐}}, \tag{12}$$

where the current per step is $J^{ss} = R_{up} - R_{down}$ with $R_{up} = R_{[i,i+1]}$ ($R_{down} = R_{[i,i-1]}$) corresponding to the occupancy rate of states moving upward (downward). The contribution due to the relative

entropy between waiting time distributions is

$$\dot{S}_{WTD} = \frac{1}{\mathcal{T}} R_{up} p_{⌐} D[\psi(t|⌐)||\psi(t|⌐)] + \frac{1}{\mathcal{T}} R_{down} p_{⌐} D[\psi(t|⌐)||\psi(t|⌐)]. \tag{13}$$

As in the previous examples, the latter term can produce a lower bound on the total entropy production rate even in the absence of observable currents, in which case $\dot{S}_{aff} = 0$. Without chemical work ($\Delta\mu = 0$), however, the waiting time distributions of the and processes become identical and the contribution of $\dot{S}_{WTD}$ vanishes as well.

Let us apply the molecular motor semi-Markov entropy production framework to a specific example. We consider the following two-state molecular motor model of a power stroke engine that works by hydrolizing ATP against an external force $F$, see Fig. 4a.

The state of the motor is described by its physical position and its internal state, which can be either active, that is, capable of hydrolyzing ATP, or passive. We label the active and passive states as $i'$ and $i$, respectively, with $i = 0, \pm 1, \pm 2, ....$. Owing to the translational symmetry in the system, all the spatial positions are essentially equivalent. The position of the motor is accessible to an external observer, whereas the two internal states $i$ and $i'$ are indistinguishable. An example of a trajectory is illustrated in Fig. 4b.

The chemical affinity $\Delta\mu$, arising from ATP hydrolysis, determines the degree of nonequilibrium in our system and biases the transitions $i' \leftrightarrow i + 1$, whereas the external force $F$ affects all the spatial transitions, regardless of the internal state. The transition rates between the two internal states are defined as $\omega_{i'i} = \omega_{ii'} = k_s$. Transition rates between passive states obey local detailed balance: $\omega_{i,i+1}/\omega_{i+1,i} = e^{\beta FL}$, where $L$ is the length of a single spatial jump. From the active state, the system can use the ATP to move upward with rates verifying local detailed balance $\omega_{i',i+1}/\omega_{i+1,i'} = e^{\beta(FL-\Delta\mu)}$.

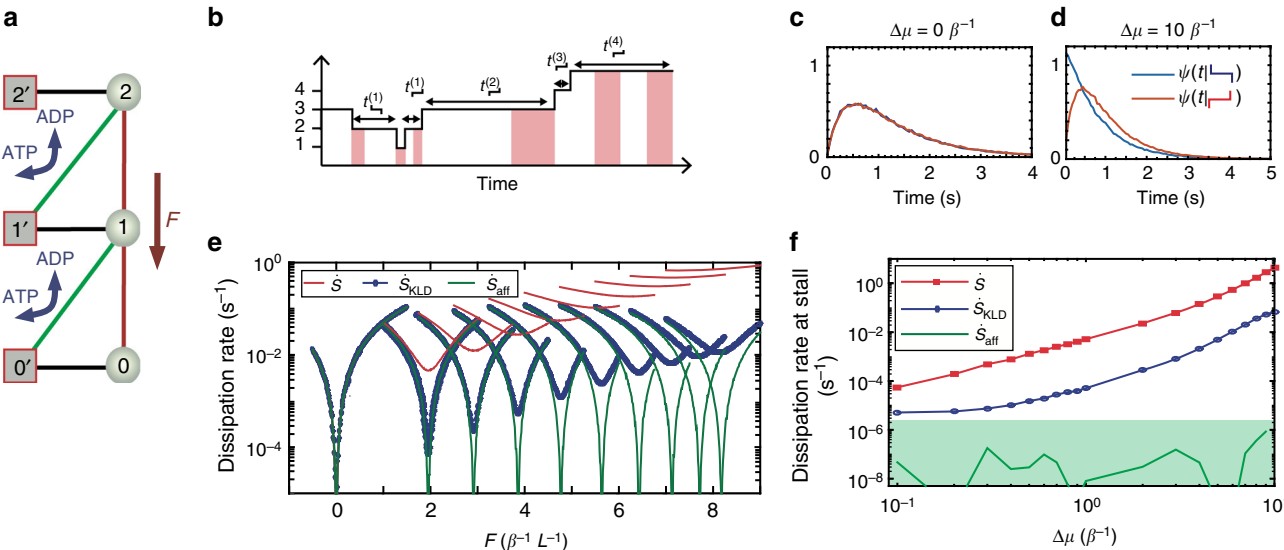

**Fig. 4** Molecular motor. **a** Illustration: Active states (red boxed squares) can use a source of chemical energy while passive states (circles) cannot. The chemical energy is used to power the motor against and external force $F$. **b** Illustration of a trajectory for four positions, where the hidden internal active state is denoted by the red shaded regions. **c, d** Waiting time distributions $\psi(t)$ for the up–up (red) and down–down (blue) transitions at stalling for $\Delta\mu = 0\ \beta^{-1}$ (**c**) and $\Delta\mu = 10\ \beta^{-1}$ (**d**). Notice that the distributions are only different in the presence of a chemical drive. **e** Total entropy production rate $\dot{S}$ (red), affinity estimator $\dot{S}_{\text{aff}}$ (green), and entropy production for semi-Markov model $\dot{S}_{\text{KLD}}$ for $\Delta\mu = 0$ (left) up to $\Delta\mu = 10$ (right), as a function of force $F$, centered at the stall force. **f** Same as (**e**) at stalling as a function of chemical drive. The affinity estimator $\dot{S}_{\text{aff}}$ offers a lower bound constrained by the statistical uncertainty due to the finite amount of data (green shaded region). Calculations were done using the parameters $k_s = 1\,\text{s}^{-1}$, $k_0 = 0.01\,\text{s}^{-1}$, and the trajectories were sampled using the Gillespie algorithm[61]

The resulting waiting time distributions are shown in Fig. 4c, d, and the estimated entropy production rates as a function of external force are depicted in Fig. 4e, with chemical potential ranging from $\Delta\mu = 0\ \beta^{-1}$ to $10\ \beta^{-1}$. The total entropy production rate $\dot{S}$ is calculated using Eq. (7). As expected, the dissipation increases with the nonequilibrium driving force, and vanishes when $\Delta\mu = FL = 0$. Notice that the affinity estimator $\dot{S}_{\text{aff}}$ does not provide a lower bound to the total entropy production rate $\dot{S}$ at stalling, as it is not statistically different from zero (Fig. 4f), and thus cannot distinguish between nonequilibrium and equilibrium processes. In contrast, the semi-Markov estimator $\dot{S}_{\text{KLD}}$, which accounts for the asymmetry of the waiting time distributions provides a nontrivial positive bound, even in the absence of observable current.

## Discussion

We have analytically derived an estimator of the total entropy production rate using the framework of semi-Markov processes. The novelty of our approach is the utilization of the waiting time distributions, which can be non-Poissonian, allowing us to unravel irreversibility in hidden degrees of freedom arising in any time-series measurement of an arbitrary experimental setup. Our estimator can thus provide a lower bound on the total entropy production rate even in the absence of observable currents. Hence, it can be applied to reveal an underlying nonequilibrium process, even if no net current, flow, or drift, are present. We stress that our method fully quantifies irreversibility. Owing to the direct link between the entropy production rate and the relative entropy between a trajectory and its time reversal, as manifested in Eq. (1), our estimator provides the best possible bound on the dissipation rate utilizing time irreversibility. One can consider utilizing other properties of the waiting time distribution to bound the entropy production, through the thermodynamics uncertainty relations[4,61,62], for example.

We have illustrated our method with two possible applications: a situation where only a subsystem is accessible to an external observer and a molecular motor whose internal degrees of freedom cannot be resolved. Using these examples, we have demonstrated the advantage of our semi-Markov estimator compared with other entropy production bounds, namely, the passive- and informed-partial entropy production rates, both of which vanish at stalling conditions.

In summary, we have developed an analytic tool that can expose irreversibility otherwise undetectable, and distinguish between equilibrium and nonequilibrium processes. This framework is completely generic and thus opens opportunities in numerous experimental scenarios by providing a new perspective for data analysis.

## Methods

**Semi-Markov processes, waiting time distributions and steady states**. A semi-Markov stochastic process is a renewal process $\alpha(t)$ with a discrete set of states $\alpha = 1, 2, …, N$. The dynamics is determined by the probability densities of transition times $\psi_{\beta\alpha}(t)$, which are defined as $\psi_{\beta\alpha}(t)\text{d}t$ being equal to the probability that the system jumps from state $\alpha$ to state $\beta$ in the time interval $[t, t + \text{d}t]$ if it arrived at site $\alpha$ at time $t = 0$. By definition $\psi_{\alpha\alpha}(t) = 0$. When the system is a particle jumping between the sites of a lattice, the semi-Markov process is also called a CTRW. For clarity, we will assume this CTRW picture, that is, the system in our discussion will be a particle jumping between sites $\alpha$.

The probability densities $\psi_{\beta\alpha}(t)$ are not normalized:

$$p_{\beta\alpha} = \int_0^\infty \psi_{\beta\alpha}(t)\text{d}t \qquad (14)$$

is the probability that, given that the particle arrived at site $\alpha$, the next jump is $\alpha \rightarrow \beta$. We will assume that the particle eventually leaves any site $\alpha$, i.e., $\sum_\beta p_{\beta\alpha} = 1$. Then

$$\psi_\alpha(t) = \sum_\beta \psi_{\beta\alpha}(t) \qquad (15)$$

is normalized and it is the probability density of the residence time at site $\alpha$. It is also called the waiting time distribution. Its average

$$\tau_\alpha = \int_0^\infty \text{d}t\, t\, \psi_\alpha(t) \qquad (16)$$

is the mean residence time or mean waiting time. We can also define the waiting time distribution conditioned on a given jump $\alpha \to \beta$,

$$\psi(t|\alpha \to \beta) = \frac{\psi_{\beta\alpha}(t)}{p_{\beta\alpha}}, \qquad (17)$$

which is normalized. The function $\psi_{\beta\alpha}(t)$ is in fact the joint probability distribution of the time $t$ and the jump $\alpha \to \beta$.

The transition probabilities $p_{\beta\alpha}$ determine a Markov chain given by the visited states $\alpha_1, \alpha_2, \alpha_3, ...,$ regardless of the times when the jumps occur. The transition matrix of this Markov chain is $\{p_{\beta\alpha}\}$ and the stationary probability distribution $R_\alpha$ verifies

$$R_\beta = \sum_\alpha p_{\beta\alpha} R_\alpha, \qquad (18)$$

i.e., the distribution $R_\alpha$ is the right eigenvector of the stochastic matrix $\{p_{\beta\alpha}\}$ with eigenvalue 1. Moreover, if the Markov chain is ergodic, then the distribution $R_\alpha$ is precisely the fraction of visits the system makes to site $\alpha$ in the stationary regime. Thus, we call $R_\alpha$ the distribution of visits.

From the distribution of visits one can easily obtain the stationary distribution of the process $\alpha(t)$,

$$\pi_\alpha = \frac{\tau_\alpha R_\alpha}{\mathcal{T}}, \qquad (19)$$

since the particle visits the state $\alpha$ a fraction of steps $R_\alpha$ and spends an average time $\tau_\alpha$ in each step. The normalization constant $\mathcal{T} \equiv \sum_\alpha R_\alpha \tau_\alpha$ is the average time per step.

The stationary current in the Markov chain from state $\alpha$ to $\beta$ is

$$J_{\beta\alpha}^{\text{ss}} = p_{\beta\alpha} R_\alpha - p_{\alpha\beta} R_\beta. \qquad (20)$$

This is in fact the current per step in the original semi-Markov system since, in an ensemble of very long trajectories, it is the net number of particles that jump from $\alpha$ to $\beta$ divided by the number of steps. Since the duration of a long stationary trajectory with $K$ steps ($K \gg 1$) is $K\mathcal{T}$, the current per unit of time is $J_{\beta\alpha}^{\text{ss}}/\mathcal{T}$. Notice that the average time per step $\mathcal{T}$ acts as a conversion factor that allows one to express currents, entropy production, etc. either as per step or as per unit of time.

**The Markovian case.** If the process $\alpha(t)$ is Markovian, then the jumps are Poissonian and transition time densities are exponential. Let $\omega_{\beta\alpha}$ be the rate of jumps from $\alpha$ to $\beta$. The mean waiting time at site $\alpha$ is the inverse of the the total outgoing rate:

$$\tau_\alpha = \frac{1}{\sum_\beta \omega_{\beta\alpha}}, \qquad (21)$$

and the waiting time distributions are

$$\psi_{\beta\alpha}(t) = \omega_{\beta\alpha} e^{-t/\tau_\alpha} \qquad \psi_\alpha(t) = \psi(t|\alpha \to \beta) = \frac{e^{-t/\tau_\alpha}}{\tau_\alpha}. \qquad (22)$$

with jump probabilities $p_{\beta\alpha} = \tau_\alpha \omega_{\beta\alpha}$. Notice that the waiting time distribution $\psi(t|\alpha \to \beta)$ does not depends on $\beta$. The distribution of visits $R_\alpha$ verifies

$$R_\beta = \sum_\alpha \tau_\alpha \omega_{\beta\alpha} R_\alpha, \qquad (23)$$

and the stationary distribution $\pi_\alpha$ obeys

$$\frac{\pi_\beta}{\tau_\beta} = \sum_\alpha \omega_{\beta\alpha} \pi_\alpha, \qquad (24)$$

which is the equation for the stationary distribution that one obtains from the master equation

$$\dot{P}_\beta(t) = \sum_\alpha \left[ \omega_{\beta\alpha} P_\alpha(t) - \omega_{\alpha\beta} P_\beta(t) \right] = \sum_\alpha \omega_{\beta\alpha} P_\alpha(t) - \frac{P_\beta(t)}{\tau_\beta}. \qquad (25)$$

**Decimation of Markov chains.** Semi-Markov processes arise in a natural way when states are removed or decimated from Markov processes with certain topologies. Consider a Markov process where two sites, 1 and 2, are connected through a closed network of states $i = 3, 4, ...$ that we want to decimate, as sketched in Fig. 2c. If the observer cannot discern between states $i = 3, 4, ...,$ the resulting three-state process with $i(t) = 1, 2, H$ is a second-order semi-Markov chain. We want to calculate the effective transition time distribution $\psi_{21}^{\text{decim}}(t)$ from state 1 to state 2 in terms of the distributions $\psi_{ij}(t)$ of the initial Markov chain. For this purpose, we have to sum over all possible paths from 1 to 2 through the decimated network.

Consider first the paths with exactly $n + 1$ jumps, like $\gamma_{n+1} = \{1 \to i_1 \to i_2 ... i_n \to 2\}$, where $i_k = 3, 4, ...$. The probability that such a path occurs with an exact

duration $t$ is

$$P(\gamma_{n+1}, t) = \int_{\sum_k t_k = t} dt_1 \, dt_2 ... \, dt_{n+1} \, \psi_{i_1,1}(t_1) \, \psi_{i_2,i_1}(t_2) ... \, \psi_{2,i_n}(t_{n+1}). \qquad (26)$$

This is a convolution. If one performs the Laplace transform on all time-dependent functions, generically denoted by a tilde,

$$\tilde{\psi}(s) \equiv \int_0^\infty dt \, e^{-st} \psi(t) \qquad (27)$$

then Eq. (26) simplifies to

$$\tilde{P}(\gamma_{n+1}, s) = \tilde{\psi}_{i_1,1}(s) \tilde{\psi}_{i_2,i_1}(s) ... \, \tilde{\psi}_{2,i_n}(s). \qquad (28)$$

The transition time distribution $\psi_{21}^{\text{decim}}(t)$ in the decimated network is the sum of $P(\gamma_{n+1}, t)$ over all possible paths with an arbitrary number of steps. For Laplace transformed distributions, this is written as

$$\tilde{\psi}_{21}^{\text{decim}}(s) = \sum_{n=0}^\infty \sum_{\{i_1,...,i_n\}} \tilde{\psi}_{i_1,1}(s) \tilde{\psi}_{i_2,i_1}(s) ... \, \tilde{\psi}_{2,i_n}(s). \qquad (29)$$

where the sum runs over all possible paths, that is, the indexes $i_k = 3, 4, ...$ take on all possible values corresponding to decimated sites. Then the sum can be expressed in terms of the matrix $\Psi(t)$ whose entries are the transition time densities $[\Psi(t)]_{ji} = \psi_{ji}(t), i, j = 3, 4, ...$. If $\tilde{\Psi}(s)$ is the corresponding Laplace transform of that matrix, one has

$$\begin{aligned} \tilde{\psi}_{21}^{\text{decim}}(s) &= \sum_{n=0}^\infty \sum_{i,j} \tilde{\psi}_{i,1}(s) \left[ \tilde{\Psi}(s)^n \right]_{ji} \tilde{\psi}_{2,j}(s) \\ &= \sum_{i,j} \tilde{\psi}_{i,1}(s) \left[ \mathbb{I} - \tilde{\Psi}(s) \right]_{ji}^{-1} \tilde{\psi}_{2,j}(s) \end{aligned} \qquad (30)$$

which is a sum only over all the decimated sites $i, j = 3, 4, ...$ that are connected to sites 1 and 2, respectively.

The decimation procedure can be used to derive transition time distributions in a kinetic network when the observer cannot discern among a set of states, say 3, 4, 5,..., that are generically labeled as $H$ for hidden, as in Fig. 2c. For the specific case of the figure, the effective transition time distribution from site 1 to site $H$, for instance, can be written as

$$\psi_{H1}^{\text{eff}}(t) = \psi_{31}(t) + \psi_{51}(t), \qquad (31)$$

whereas the distributions for jumps starting at $H$ depend on the previous state. For instance, if $H$ is reached from 1, the random walk within $H$ starts at site 3 with probability $p_{31}/(p_{31} + p_{51})$ and site 5 with probability $p_{51}/(p_{31} + p_{51})$. The transition time distribution corresponding to the jump $[1H] \to [H2]$ is

$$\psi_{[H2]\leftarrow[1H]}^{\text{eff}}(s) = \frac{p_{31}}{p_{31} + p_{51}} \left[ \mathbb{I} - \tilde{\Psi}(s) \right]_{43}^{-1} \tilde{\psi}_{24}(s) + \frac{p_{51}}{p_{31} + p_{51}} \left[ \mathbb{I} - \tilde{\Psi}(s) \right]_{45}^{-1} \tilde{\psi}_{24}(s) \qquad (32)$$

where the matrix $\tilde{\Psi}(s)$ is a $3 \times 3$ matrix corresponding to the Laplace transform of the transition time distributions among sites 3, 4, and 5.

**Irreversibility in semi-Markov processes.** Here we calculate the relative entropy between a stationary trajectory $\gamma$ and its time reversal $\tilde{\gamma}$ in a generic semi-Markov process. A trajectory $\gamma$ is fully described by the sequence of jumps (see Fig. 5):

$$\gamma = \{(\alpha_1 \to \alpha_2, t_1), (\alpha_2 \to \alpha_3, t_2), ..., (\alpha_{n-1} \to \alpha_n, t_{n-1}), (\alpha_n \to \alpha_{n+1}, t_n)\} \qquad (33)$$

and occurs with a probability (conditioned on the initial jump $\alpha_0 \to \alpha_1$ at $t = 0$)

$$P(\gamma) = \psi_{\alpha_2,\alpha_1}(t_1) \psi_{\alpha_3,\alpha_2}(t_2) ... \, \psi_{\alpha_{n+1},\alpha_n}(t_n). \qquad (34)$$

The reverse trajectory is

$$\tilde{\gamma} = \{(\tilde{\alpha}_n \to \tilde{\alpha}_{n-1}, t_n), ..., (\tilde{\alpha}_2 \to \tilde{\alpha}_1, t_2), (\tilde{\alpha}_1 \to \tilde{\alpha}_0, t_1)\}, \qquad (35)$$

where we assume, for the sake of generality, that states can change under time reversal, $\tilde{\alpha}$ being the time reversal of state $\alpha$. The probability to observe $\tilde{\gamma}$, conditioned on the initial jump $\tilde{\alpha}_{n+1} \to \tilde{\alpha}_n$ at $t = 0$, is

$$P(\tilde{\gamma}) = \psi_{\tilde{\alpha}_0,\tilde{\alpha}_1}(t_1) \psi_{\tilde{\alpha}_1,\tilde{\alpha}_2}(t_2) ... \, \psi_{\tilde{\alpha}_{n-1},\tilde{\alpha}_n}(t_n). \qquad (36)$$

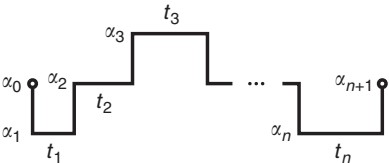

**Fig. 5** A trajectory $\gamma$ of a semi-Markov process

It is again convenient to consider the forward and backward trajectories without the waiting times, i.e.,

$$\sigma = \{\alpha_1, \alpha_2, \alpha_3, \ldots, \alpha_n, \alpha_{n+1}\} \tag{37}$$

$$\tilde{\sigma} = \{\tilde{\alpha}_n, \tilde{\alpha}_{n-1}, \ldots, \tilde{\alpha}_2, \tilde{\alpha}_1, \tilde{\alpha}_0\}, \tag{38}$$

and the probability to observe those trajectories are

$$P(\sigma) = p_{\alpha_2, \alpha_1} P_{\alpha_3, \alpha_2} \cdots p_{\alpha_{n+1}, \alpha_n} \tag{39}$$

$$P(\tilde{\sigma}) = p_{\tilde{\alpha}_0, \tilde{\alpha}_1} P_{\tilde{\alpha}_1, \tilde{\alpha}_2} \cdots p_{\tilde{\alpha}_{n-1}, \tilde{\alpha}_n} \tag{40}$$

The initial jumps of $\gamma$ and $\tilde{\gamma}$ do not contribute to the entropy production in the stationary regime. Then the relative entropy per jump reads

$$\delta S_{\mathrm{KLD}} = \lim_{n \to \infty} \frac{1}{n} \sum_\gamma P(\gamma) \ln \frac{P(\gamma)}{P(\tilde{\gamma})} = \lim_{n \to \infty} \frac{1}{n} \sum_\sigma \int_0^\infty \mathrm{d}t_1 \ldots \int_0^\infty \mathrm{d}t_n\, \psi_{\alpha_2, \alpha_1}(t_1) \ldots$$
$$\psi_{\alpha_{n+1}, \alpha_n}(t_n) \left[ \ln \frac{\psi_{\alpha_2, \alpha_1}(t_1)}{\psi_{\tilde{\alpha}_0, \tilde{\alpha}_1}(t_1)} + \ldots + \ln \frac{\psi_{\alpha_{n+1}, \alpha_n}(t_n)}{\psi_{\tilde{\alpha}_{n-1}, \tilde{\alpha}_n}(t_n)} \right]. \tag{41}$$

Each time integral can be written as

$$\int_0^\infty \mathrm{d}t\, \psi_{\mu\beta}(t) \ln \frac{\psi_{\mu\beta}(t)}{\psi_{\tilde{\alpha}\tilde{\beta}}(t)} = p_{\mu\beta} \ln \frac{p_{\mu\beta}}{p_{\tilde{\alpha}\tilde{\beta}}} + p_{\mu\beta} D\left[ \psi(t|\beta \to \mu) || \psi(t|\tilde{\beta} \to \tilde{\alpha}) \right], \tag{42}$$

where $\alpha, \beta, \mu$ is a substring of the forward trajectory $\sigma$ ($\alpha = \alpha_k, \beta = \alpha_{k+1}, \mu = \alpha_{k+2}$). Inserting this expression in Eq. (41),

$$\delta S_{\mathrm{KLD}} = \lim_{n \to \infty} \frac{1}{n} D[P(\sigma) || P(\tilde{\sigma})] + \sum_{\alpha\beta\mu} p_{\mu\beta} p_{\beta\alpha} R_\alpha D\left[ \psi(t|\beta \to \mu) || \psi(t|\tilde{\beta} \to \tilde{\alpha}) \right]$$
$$= \sum_{\alpha\beta} p_{\beta\alpha} R_\alpha \ln \frac{p_{\beta\alpha}}{p_{\tilde{\alpha}\tilde{\beta}}} + \sum_{\alpha\beta\mu} p_{\mu\beta} p_{\beta\alpha} R_\alpha D\left[ \psi(t|\beta \to \mu) || \psi(t|\tilde{\beta} \to \tilde{\alpha}) \right]. \tag{43}$$

Notice that $p_{\beta\alpha} R_\alpha$ is the probability to observe the sequence $\alpha, \beta$ in the stationary forward trajectory and $p_{\mu\beta} p_{\beta\alpha} R_\alpha$ is the probability to observe the sequence $\alpha, \beta, \mu$. Finally, we can obtain the expression used in the main text for the entropy production per unit of time dividing by the conversion factor $\mathcal{T}$ (average time per step), that is $\dot{S} = \delta S / \mathcal{T}$. The result is

$$\dot{S}_{\mathrm{KLD}} = \dot{S}_{\mathrm{aff}} + \dot{S}_{\mathrm{WTD}}, \tag{44}$$

where the entropy production corresponding to the affinity of states reads

$$\dot{S}_{\mathrm{aff}} = \frac{1}{\mathcal{T}} \sum_{\alpha\beta} p_{\beta\alpha} R_\alpha \ln \frac{p_{\beta\alpha}}{p_{\tilde{\alpha}\tilde{\beta}}}, \tag{45}$$

and the one corresponding to the waiting time distributions is

$$\dot{S}_{\mathrm{WTD}} = \frac{1}{\mathcal{T}} \sum_{\alpha\beta\mu} p_{\mu\beta} p_{\beta\alpha} R_\alpha D\left[ \psi(t|\beta \to \mu) || \psi(t|\tilde{\beta} \to \tilde{\alpha}) \right]. \tag{46}$$

If $\alpha = \tilde{\alpha}$, then the affinity entropy production can be written as

$$\dot{S}_{\mathrm{aff}} = \frac{1}{\mathcal{T}} \sum_{\alpha < \beta} J_{\beta\alpha}^{\mathrm{ss}} \ln \frac{p_{\beta\alpha}}{p_{\alpha\beta}}, \tag{47}$$

which vanishes in the absence of currents.

**Second-order semi-Markov processes.** A 2nd-order semi-Markov process $i(t)$ also describes the trajectory of a system that jumps among a discrete set of states $i = 1, 2, \ldots$. However, $i(t)$ is not semi-Markov because the transition time distributions depend on the previous state $i_{\mathrm{prev}}(t)$ visited right before the last jump. Hence, the vector $\alpha(t) \equiv [i_{\mathrm{prev}}(t)\, i(t)]$ is indeed a semi-Markov process.

To quantify the irreversibility of a second-order Markov chain, we introduce the time-reversal state of $\alpha = [ij]$, which is $\tilde{\alpha} = [ji]$. However, this is not enough to reconstruct the backward trajectory, since there is a shift compared with the simple semi-Markov case, as illustrated in Fig. 6. In the forward trajectory, the system spends a time $t_k$ in state $\alpha_k = [i_{k-1} i_k]$, with $k = 1, \ldots, n$, whereas in the backward trajectory it spends the same time $t_k$ in state $\tilde{\alpha}_{k+1} = [i_{k+1} i_k]$. Consequently, the probabilities of the forward and backward trajectories are, respectively,

$$P(\gamma) = \psi_{\alpha_2, \alpha_1}(t_1) \psi_{\alpha_3, \alpha_2}(t_2) \cdots \psi_{\alpha_{n+1}, \alpha_n}(t_n) \tag{48}$$

$$P(\tilde{\gamma}) = \psi_{\tilde{\alpha}_1, \tilde{\alpha}_2}(t_1) \psi_{\tilde{\alpha}_2, \tilde{\alpha}_3}(t_2) \cdots \psi_{\tilde{\alpha}_n, \tilde{\alpha}_{n+1}}(t_n). \tag{49}$$

Repeating the arguments of the previous section, one obtains

$$\delta S_{\mathrm{KLD}} = \sum_{\alpha\beta} p_{\beta\alpha} R_\alpha \ln \frac{p_{\beta\alpha}}{p_{\tilde{\alpha}\tilde{\beta}}} + \sum_{\alpha\beta} p_{\beta\alpha} R_\alpha D\left[ \psi(t|\alpha \to \beta) || \psi(t|\tilde{\beta} \to \tilde{\alpha}) \right]. \tag{50}$$

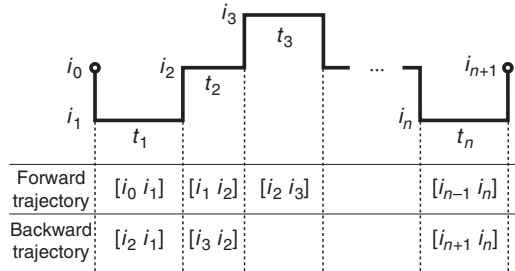

**Fig. 6** A trajectory $\gamma$ of a second-order semi-Markov process

The contribution to the entropy production (per step) due to the state affinities now reads

$$\dot{S}_{\mathrm{aff}} = \frac{1}{\mathcal{T}} \sum_{i,j,k} R_{[ij]} p([ij] \to [jk]) \ln \frac{p([ij] \to [jk])}{p([kj] \to [ji])}$$
$$= \frac{1}{\mathcal{T}} \sum_{i,j,k} p(ijk) \ln \frac{p([ij] \to [jk])}{p([kj] \to [ji])}, \tag{51}$$

and the contribution due to the waiting time distributions is given by

$$\dot{S}_{\mathrm{WTD}} = \frac{1}{\mathcal{T}} \sum_{i,j,k} R_{[ij]} p([ij] \to [jk]) D[\psi(t|[ij] \to [jk]) || \psi(t|[kj] \to [ji])]$$
$$= \frac{1}{\mathcal{T}} \sum_{i,j,k} p(ijk) D[\psi(t|[ij] \to [jk]) || \psi(t|[kj] \to [ji])], \tag{52}$$

where $p(ijk) = R_{[ij]} p([ij] \to [jk])$ is the probability to observe the sequence $i \to j \to k$ in the trajectory and $p(ij) = R_{[ij]}$ is the probability to observe the sequence $i \to j$.

It is interesting to particularize Eq. (51) to a ring with $N$ sites. This is the case of our examples—the hidden network and the molecular motor. In this case, in the stationary regime,

$$p(ijk) - p(kji) = p(ijk) + p(iji) - p(iji) - p(kji)$$
$$= p(ij) - p(ji) = J^{\mathrm{ss}} \tag{53}$$

since each site has only two neighbors and therefore $p(ijk) + p(iji) = p(ij)$ for any triplet of contiguous sites $ijk$. Here $J^{\mathrm{ss}}$ is the stationary current between any pair of contiguous sites. Hence, we can write the affinity as

$$\dot{S}_{\mathrm{aff}} = \frac{1}{\mathcal{T}} \sum_{i<j<k} [p(ijk) - p(kji)] \ln \frac{p([ij] \to [jk])}{p([kj] \to [ji])}$$
$$= \frac{J^{\mathrm{ss}}}{\mathcal{T}} \ln \frac{p([1,2] \to [2,3]) \ldots p([N-1,N] \to [N,1]) p([N,1] \to [1,2])}{p([1,N] \to [N,N-1]) \ldots p([3,2] \to [2,1]) p([2,1] \to [1,N])} \tag{54}$$

which is proportional to the current. The argument of the logarithm also vanishes at zero current (see Eq. (57) below); consequently, the affinity entropy tends to zero quadratically when as the force is tuned to the stalling condition. This is the usual behavior in linear irreversible thermodynamics, but recall that for semi-Markov processes the affinity entropy production misses the nonequilibrium signature that is present in the waiting time distributions and is assessed by $\dot{S}_{\mathrm{WTD}}$.

**Affinity and informed partial entropy production.** Here we show that the informed partial entropy production equals the affinity entropy production for the case of a generic hidden kinetic network proposed in the main text where the observed network forms a single cycle.

First, let us generalize the detailed balance condition for a second-order Markov ring with three states, $i = 1, 2, H$, and zero stationary current. The stationary distribution $R_{[ij]}$ verifies the master Eq. (18):

$$R_{[12]} = R_{[H1]} p([H1] \to [12]) + R_{[21]} p([21] \to [12])$$
$$R_{[2H]} = R_{[12]} p([12] \to [2H]) + R_{[H2]} p([H2] \to [2H]) \tag{55}$$
$$R_{[H1]} = R_{[2H]} p([2H] \to [H1]) + R_{[1H]} p([1H] \to [H1]).$$

If the current vanishes, $R_{[ij]} = R_{[ji]}$ for all $i, j$, and these equations reduce to

$$R_{[12]} p([21] \to [1H]) = R_{[H1]} p([H1] \to [12])$$
$$R_{[2H]} p([H2] \to [21]) = R_{[12]} p([12] \to [2H]) \tag{56}$$
$$R_{[H1]} p([1H] \to [H2]) = R_{[2H]} p([2H] \to [H1]).$$

Multiplying the three equations we get the generalized detailed balance condition:

$$J^{\mathrm{ss}} = 0 \Rightarrow p([1H] \to [H2]) p([H2] \to [21]) p([21] \to [1H])$$
$$= p([2H] \to [H1]) p([H1] \to [12]) p([12] \to [2H]). \tag{57}$$

In the observable network, the transitions from states 1 and 2 are still Poissonian and independent of the previous state:

$$p([H2] \to [21]) = \tau_2 \omega_{12} \qquad p([21] \to [1H]) = \tau_1 \omega_{H1}$$
$$p([H1] \to [12]) = \tau_1 \omega_{21} \qquad p([12] \to [2H]) = \tau_2 \omega_{H2}. \tag{58}$$

At stall force, the generalized detailed balance condition in Eq. (57) holds and can be written as

$$p([1H] \rightarrow [H2])\omega_{12}^{\text{stall}}\omega_{H1} = p([2H] \rightarrow [H1])\omega_{21}^{\text{stall}}\omega_{H2}, \tag{59}$$

where we have taken into account that only the rates $\omega_{12}$ and $\omega_{21}$ are tuned in the protocol proposed by Polettini and Esposito to obtain the informed partial entropy production.

The current in the direction $1 \rightarrow 2 \rightarrow H \rightarrow 1$ in the stationary regime can be written as $J^{\text{ss}}/T = \omega_{12}\pi_2 - \omega_{21}\pi_1$. Then, at stall force $\omega_{12}^{\text{stall}}\pi_2^{\text{stall}} = \omega_{21}^{\text{stall}}\pi_1^{\text{stall}}$. With all these considerations, the argument of the logarithm in Eq. (9) of the main text can be written as

$$
\begin{aligned}
\frac{\omega_{12}\,\pi_2^{\text{stall}}}{\omega_{21}\,\pi_1^{\text{stall}}} &= \frac{\omega_{12}\,\omega_{21}^{\text{stall}}}{\omega_{21}\,\omega_{12}^{\text{stall}}} = \frac{\omega_{12}\,p([1H] \rightarrow [H2])\omega_{H1}}{\omega_{21}\,p([2H] \rightarrow [H1])\omega_{H2}} \\
&= \frac{p([1H] \rightarrow [H2])p([H2] \rightarrow [21])p([21] \rightarrow [1H])}{p([2H] \rightarrow [H1])p([H1] \rightarrow [12])p([12] \rightarrow [2H])}.
\end{aligned}
\tag{60}
$$

Comparing Eq. (9) in the main text with Eq. (54), one immediately gets $\dot{S}_{\text{IP}} = \dot{S}_{\text{aff}}$.

## Data availability

The data that support the findings of this study are available from the corresponding author upon reasonable request.

## Code availability

Source code is available from the corresponding authors upon reasonable request.

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

## Acknowledgements

I.A.M. and J.M.R.P. acknowledge funding from the Spanish Government through grants TerMic (FIS2014-52486-R) and Contract (FIS2017-83709-R). I.A.M. acknowledges funding from Juan de la Cierva program. G.B. acknowledges the Zuckerman STEM Leadership Program. J.M.H. is supported by the Gordon and Betty Moore Foundation as a Physics of Living Systems Fellow through Grant No. GBMF4513.

## Author contributions

J.M.R.P. conceived the project. I.A.M. and G.B. performed the numerical simulations and analyzed the data. All authors discussed the results and wrote the paper.

## Additional information

**Competing interests:** The authors declare no competing interests.

