## [peer review file · Nature Communications]

Reviewer #1 (Remarks to the Author):

Martinez et al consider a general class of non-equilibrium systems with hidden states. Such models are relevant for biological systems such as molecular motors. The key question is how to estimate the entropy production rate (EPR) from partial observations of the stochastic dynamics of such systems. To this end, various partial entropy production measures have been proposed in prior work that are carefully discussed by the authors. Martinez and colleagues discuss how these hidden network models are described by semi-Markov processes. The key result of the paper is a new method based on the Kullback-Leibler Divergence that not only captures the usual affinity entropy production, but also a newly identified contribution due to the waiting time distribution.

An interesting feature of the new and improved measure for the EPR proposed by the authors, is that it can also detect non-zero EPR in the absence of probability currents. To illustrate their approach, the authors consider various examples, including a model for a molecular motor near stalling conditions. The method appears to be natural and generalizable to various systems.

The paper is well-written and appears to be physically solid. This work makes a significant contribution to the ongoing debate on accurate entropy production measures. I also appreciate it that the authors illustrate their ideas by some relevant examples such as a molecular motor. However, because of several concerns that I listed below, I cannot recommend publication of the manuscript in its current form.

1. To make a case for the broad readership of nature communications, I think the authors need to discuss what we could learn about a system (such a molecular motor) by estimating the entropy production, especially since the estimate is likely to be a weak lower bound.
2. Why is it so important to measure the EPR near stalling? It seems like the issue could be resolved by simply estimating EPR contributions under non-stalling conditions.
3. For some cases the authors prove a hierarchy between various estimates of the EPRs used so far and prove that their new measure does the best job for these examples. The question is whether it's the ultimate solution, or is it possible that one can do even better (for the hidden networks they consider)? This needs to be discussed more clearly in the text.
4. Minor point: It should be stated in the captions or figures what units are used.

Reviewer #2 (Remarks to the Author):

This paper contains interesting results, and at the same time it is not clear on all its points. This has to do with the fact that I do not consider myself an expert in the sub-field of entropy production

thus hopefully my review could be used constructively to slightly improve the presentation for non-experts or completely ignored.

The thing I liked about the paper is its main equations, which give the KLD

entropy production rate as a sum of two terms Eq. 3 and Eq. 4. When currents are absent S_{aff} eq. 3 is zero, but still the rate S_{WTD} is non vanishing, and thus a novel aspect of the approach is that it can be used to analyse entropy production in the absence of currents. This is a major claim of the paper, which should be of interest to the community.

At the same time I found some of the statements in the paper, especially the motivation slightly overwhelming. For example the authors motivate the paper with CTRW claiming correctly that this model has to do with popular biological experiments. Let us examine the three references cited:

1) Ref. [30] on chromosome dynamics. The conclusion in [30] is that fBM is the valid model not CTRW, so this reference actually works against the authors.

2) Anomalous diffusion in the Plasma membrane [31]. Saxton's paper is theoretical. See paper from Diego Krapf in PNAS which shows that CTRW is

non-ergodic, in this context. It is my view that Saxton's paper, already points out to trapping models, and this has been strengthened in recent experiments, which show that the processes are non ergodic (and hence the conclusion of the paper by this ideology are simply not relevant).

3) Ref [32] is again theoretical, I see no connection to real life.

There is a vast literature on CTRW in the cell environment, but nearly all of it shows that the process are non-stationary (see PCCP review of Metzler et al,

for example). This implies that statements of generality seem to me a bit too strong, since in the cell rates change from one cell to the other, the cell lives once, the cell is non stationary etc. Of course this does not imply that the paper is dealing with an unimportant problem, I fully support the work, but not the claims of general validity. To put differently the authors should explain better the many ways this work should be generalized: fractional Brownian motion, ageing CTRW etc. Then with respect to determination of hidden networks, there is a vast literature on this for example in the context of CTRW, see Klafter old work on single molecule dynamics, in the context

of single ion channels the literature is really gigantic. It will be easy for the authors to read the relevant literature and fix this minor point. I also note work of M. Fisher and A. Kolomeisky on CTRW dynamics for Kinesin molecules that seem to me relevant to the current paper, as well as work

of Nature physics of Kindermann, Dechant et on single atom diffusion

in periodic potentials (2016).

However, the main issues I am asking the authors to clarify a bit better, for the sake of the general readership are the following:

a) What is entropy in a general non-equilibrium setting? Note that to my mind there are many definitions of \dot{S} , thermodynamics is one option,

but this is clearly not relevant, Shannon entropy, or Kolmogorov entropy etc?

b) Are we dealing with definition physics? for example we have \dot{S} greater than \dot{S}_{KLD} and then one finds a nice expression for the latter. So let

us say that \dot{S}_{KLD} can be evaluated in the lab. But can \dot{S} be evaluated? If so for models in paper how well is the bound working?

If not why is a bound on a non computable observable (non measurable)

worthy? for example Kolmogorov entropy rate cannot be evaluated with a finite machine for chaotic systems as well known.

c) then in equation 7 the authors give the rate of entropy production.

It actually confused me a lot (again I am not an expert) since if we have the rate of entropy production why at all bother with the bound. Or why not

delegate the paper to special journals. Yes Eq. 7 is a steady state statement, and maybe this is why it is limited compared to the bound, but not so

clear to me why we should go beyond the steady state/.

Another technical issue, is the following. The authors define a forward and backward path. If we have a path going in discrete steps in time, between a few states, we can expect to find the direct path and then its reverse in experimental situation. For CTRW we have also the times in all the states.

Hence we have the condition given in the text $t = \sum t_n$. If I understood correctly, one must look for the reverse path, that follows the same time intervals as the forward one. This seems to me a path of measure zero.

It is unlikely to be found, so how do the authors see the real evaluation

of reverse trajectories in continuous time. In any experiment the waiting times fluctuate, indeed that is an important new part of the model,

and hence reconstruction of a reverse path seems to me unlikely.

To conclude as a theoretical paper I find the main equation in the paper novel and interesting. The entropy rate can be decomposed into two parts one vanishes for zero current. Hopefully the authors can use this report in constructive manner, at least that was the main intention of this report.

Reviewer #3 (Remarks to the Author):

Report on the manuscript NCOMMS-19-01647 Title: Inferring broken detailed balance in the absence of observable currents Authors: I. A. Martínez, G. Bisker, J. M. Horowitz, and J. M. R. Parrondo For general semi-Markov processes, the Authors are showing in this manuscript that the Kullback-Leibler divergence between the probabilities to observe a trajectory and the time-reversed trajectory has two non-negative contributions: (1) the well-known contribution from the transitions between the states of the stochastic process and (2) another contribution coming from the waiting time distributions of the process. Consistently, the semi-Markov process is reversible if and only if the two contributions are equal to zero. The contribution (2) is a new result of great importance for the study of irreversibility in many different contexts. In the absence of observable currents, the standard contribution (1) can be vanishing, so that the contribution (2) may still provide a lower bound on the thermodynamic entropy production, if this latter is related to the Kullback-Leibler divergence. The manuscript is clearly written and address a broad audience. By the importance of the experimental relevance and the novelty of the result, this manuscript clearly deserves publication in Nature Communications. There are a few typos to be fixed, e.g., "An observer unable resolve..." → "An observer unable to resolve..."; "In order to asses..." → "In order to assess..."; "less then..." → "less than..."; there is a weird symbol for the identity matrix in (B5) and (B7).

Reviewers' comments:

Reviewer #1 (Remarks to the Author):

Martinez et al consider a general class of non-equilibrium systems with hidden states. Such models are relevant for biological systems such as molecular motors. The key question is how to estimate the entropy production rate (EPR) from partial observations of the stochastic dynamics of such systems. To this end, various partial entropy production measures have been proposed in prior work that are carefully discussed by the authors. Martinez and colleagues discuss how these hidden network models are described by semi-Markov processes. The key result of the paper is a new method based on the Kullback-Leibler Divergence that not only captures the usual affinity entropy production, but also a newly identified contribution due to the waiting time distribution.

An interesting feature of the new and improved measure for the EPR proposed by the authors, is that it can also detect non-zero EPR in the absence of probability currents. To illustrate their approach, the authors consider various examples, including a model for a molecular motor near stalling conditions. The method appears to be natural and generalizable to various systems.

The paper is well-written and appears to be physically solid. This work makes a significant contribution to the ongoing debate on accurate entropy production measures. I also appreciate it that the authors illustrate their ideas by some relevant examples such as a molecular motor. However, because of several concerns that I listed below, I cannot recommend publication of the manuscript in its current form.

Authors' response: We thank the reviewer for the kind words and strong support, as well and the time an effort invested in the review process. We are happy to address all the points raised in the review.

To make a case for the broad readership of nature communications, I think the authors need to discuss what we could learn about a system (such a molecular motor) by estimating the entropy production, especially since the estimate is likely to be a weak lower bound.

*Authors' response: We agree that such a discussion would render the paper more accessible to the broad readership of *Nature Communications*. We have stated in the abstract that "Detecting dissipation is essential for our basic understanding of the underlying physical mechanism...", and we have now elaborated by adding the following paragraphs to the introduction, in the beginning:*

Irreversibility is the telltale sign of nonequilibrium dissipation [1, 2]. Systems operating far-from-equilibrium utilize part of their free energy budget to perform work, while the rest is dissipated into the environment. Estimating the amount of free energy lost to dissipation is mandatory for a complete energetics characterization of such physical systems. For example, it is essential for understanding the underlying mechanism and efficiency of natural Brownian engines, such as RNA-polymerases or kinesin molecular motors, or for optimizing the performance of artificial devices [3-5].

Often, the manifestation of nonequilibrium dissipation is quite dramatic...

And in the end of the introduction:

In addition, our quantitative lower bound on the entropy production rate can be used to shed light on the efficiency of molecular motors operation and on the entropic cost of maintaining their far-from-equilibrium dynamics [40-42].

Why is it so important to measure the EPR near stalling? It seems like the issue could be resolved by simply estimating EPR contributions under non-stalling conditions.

Authors' response: Basically, the goal was not to force a physical system into stalling, but rather show that our estimator can detect entropy production even in stalling conditions in the absence of observable currents.

In order to make this point clear, we have added the following sentence to the Conclusions section:

"Our estimator can thus provide a lower bound on the total entropy production rate even in the absence of observable currents. Hence, it can be applied to reveal an underlying nonequilibrium process, even if no net current, flow, or drift, are present."

For some cases the authors prove a hierarchy between various estimates of the EPRs used so far and prove that their new measure does the best job for these examples. The question is whether it's the ultimate solution, or is it possible that one can do even better (for the hidden networks they consider)? This needs to be discussed more clearly in the text.

Authors' response: As we have pointed out in the conclusion section:

"Owing to the direct link between the entropy production rate and the relative entropy between a trajectory and its time-reversal, as manifested in Eq. (1), our estimator provides the best possible bound on the dissipation rate utilizing time-irreversibility."

In order to entertain the idea of better bounds, we have added the following sentence:

"One can consider utilizing other properties of the waiting time distributions to bound the entropy production, through the thermodynamic uncertainty relations [60, 60], for example."

Minor point: It should be stated in the captions or figures what units are used.

Authors' response: We thank the reviewer for pointing this one. The dissipation rate in the figures are in units of $[s^{-1}]$, since we have taken the Boltzmann constant to be one, and the force F and the chemical potential $\Delta\mu$ are presented in the units of $(\beta L)^{-1}$ and β^{-1} respectively, where β is the inverse temperature and L is a characteristic length scale in the problem. We have corrected the corresponding axis labels.

Reviewer #2 (Remarks to the Author):

This paper contains interesting results, and at the same time it is not clear on all its points. This has to do with the fact that I do not consider myself an expert in the sub-field of entropy production thus hopefully my review could be used constructively to slightly improve the presentation for non-experts or completely ignored.

Authors' response: We thank the reviewer for the support, and for time and effort invested in the review process. We are happy to address all the points raised and to clarify our results.

The thing I liked about the paper is its main equations, which give the KLD entropy production rate as a sum of two terms Eq. 3 and Eq. 4. When currents are absent S_{aff} eq. 3 is zero, but still the rate S_{WTD} is non vanishing, and thus a novel aspect of the approach is that it can be used to analyze entropy production in the absence of currents. This is a major claim of the paper, which should be of interest to the community.

Authors' response: We thank the reviewer for the kind words.

At the same time, I found some of the statements in the paper, especially the motivation slightly over whelming. For example, the authors motivate the paper with CTRW claiming correctly that this model has to do with popular biological experiments. Let us examine the three references cited:

1) Ref. [30] on chromosome dynamics. The conclusion in [30] is that fBM is the valid model not CTRW, so this reference actually works against the authors.

2) Anomalous diffusion in the Plasma membrane [31]. Saxton's paper is theoretical. See paper from Diego Krapf in PNAS which shows that CTRW is non-ergodic, in this context. It is my view that Saxton's paper, already points out to trapping models, and this has been strengthened in recent experiments, which show that the processes are non-ergodic (and hence the conclusion of the paper by this ideology are simply not relevant).

3) Ref [32] is again theoretical, I see no connection to real life. There is a vast literature on CTRW in the cell environment, but nearly all of it shows that the process are non-stationary (see PCCP review of Metzler et al, for example). This implies that statements of generality seem to me a bit too strong, since in the cell rates change from one cell to the other, the cell lives once, the cell is non stationary etc.

Of course this does not imply that the paper is dealing with an unimportant problem, I fully support the work, but not the claims of general validity. To put differently the authors should explain better the many ways this work should be generalized: fractional Brownian motion, ageing CTRW etc. Then with respect to determination of hidden networks, there is a vast literature on this for example in the context of CTRW, see Klafter old work on single molecule dynamics, in the context of single ion channels the literature is really gigantic. It will be easy for the authors to read the relevant literature and fix this minor point. I also note work of M. Fisher and A. Kolomeisky on CTRW dynamics for Kinesin molecules that seem to me relevant to the current paper, as well as work of Nature physics of Kindermann, Dechant et on single atom diffusion in periodic potentials (2016).

Authors' response: We acknowledge the referee for the thorough analysis of our references. Our main point was to emphasize the emergence of CTRW dynamics for partially observed Markovian systems, rather than claiming to what extent CTRW is actually a good predictive model. Following the valuable suggestions of the referee, we have removed the inappropriate references, and included the new references, as recommended. We have revised the relevant paragraph in the introduction, which now reads:

Such models emerge in a plethora of contexts [34-36] ranging from economy and finance [37] to biology, as in the case of kinesin dynamics [38] or in the anomalous diffusion of the Kv2.1 potassium channel [39]. In fact, as we show below...

Regarding the other issue raised by the referee, we agree that the processes in the cell are in principle non-ergodic and non-stationary. However, if we choose observation times that are short enough, we can assume stationarity and ergodicity, or we can consider *in vitro* experiments under controlled conditions. Further generalization such as fractional Brownian motion and ageing CTRW are beyond the scope of this work.

What is entropy in a general non-equilibrium setting? Note that to my mind there are many definitions of \dot{S} , thermodynamics is one option, but this is clearly not relevant, Shannon entropy, or Kolmogorov entropy etc?

Are we dealing with definition physics? for example we have \dot{S} greater than \dot{S}_{KLD} and then one finds a nice expression for the latter. So let us say that \dot{S}_{KLD} can be evaluated in the lab. But can \dot{S} be evaluated? If so for models in paper how well is the bound working? If not why is a bound on a non-computable observable (non measurable) worthy? for example Kolmogorov entropy rate cannot be evaluated with a finite machine for chaotic systems as well known.

Authors' response: We thank the reviewer for raising these questions, and we agree that a more pedagogical approach would make the paper clearer and more accessible. We have thus added the following paragraphs to the introduction:

"Our understanding of the connection between irreversibility and dissipation has deepened in recent years with the formulation of stochastic thermodynamics, which has been verified in numerous experiments on meso-scale systems [19–22]. Within this framework, it is possible to evaluate quantities as the entropy along single non-equilibrium trajectories [23]. [...]

The entropy production \dot{S} in Eq. (1) has a clear physical meaning. It is the usual entropy production defined in irreversible thermodynamics by assuming that the reservoirs surrounding the system are in equilibrium. For instance, in the case of isothermal molecular motors hydrolyzing ATP to ADP+P at temperature T , the entropy production in Eq. (1) is $\dot{S} = r\Delta\mu/T - \dot{W}/T$, where r is the ATP consumption rate, $\Delta\mu = \mu_{ATP} - \mu_{ADP} - \mu_P$ is the difference between the ATP, and the ADP and P chemical potentials, and \dot{W} is the power of the motor. In many experiments, all these quantities can be measured except the rate r . Therefore, the techniques that we develop in this paper can help to estimate the ATP consumption rate, even at stalling conditions."

then in equation 7 the authors give the rate of entropy production. It actually confused me a lot (again I am not an expert) since if we have the rate of entropy production why at all bother with the bound. Or why not delegate the paper to special journals. Yes Eq. 7 is a steady state statement, and maybe this is why it is limited compared to the bound, but not so clear to me why we should go beyond the steady state.

Authors' response: We thank the reviewer for this question. Indeed Eq. (7) is the steady state entropy production rate, however, it includes a sum ($\sum_{i<j} \dots$) over all states and transitions in the system. Thus, in order to compute it, one must have access to all the microstates and the full dynamics of the system, which is impractical in realistic scenarios as we have stated in the introduction. Hence, obtaining lower bounds is sometimes the only option. In order to emphasize this point, we have added the following sentence following equation (7):

“In order to calculate the total entropy production \dot{S} according to Eq. (7), full knowledge of the steady state probability distribution $\{\pi_i\}$ and the transition rates between all the microstates $\{\omega_{ij}\}$ is required. We would like to assign a partial entropy production rate when one only has access to a limited set of states and transitions...”

Another technical issue, is the following. The authors define a forward and backward path. If we have a path going in discrete steps in time, between a few states, we can expect to find the direct path and then its reverse in experimental situation. For CTRW we have also the times in all the states. Hence we have the condition given in the text $t = \sum t_n$. If I understood correctly, one must look for the reverse path, that follows the same time intervals as the forward one. This seems to me a path of measure zero. It is unlikely to be found, so how do the authors see the real evaluation of reverse trajectories in continuous time. In any experiment the waiting times fluctuate, indeed that is an important new part of the model, and hence reconstruction of a reverse path seems to me unlikely.

Authors' response: We thank the referee for raising this question. Indeed, the time-reversal trajectory is defined by visiting all the states in reversed order with identical waiting times as explained in the main text before equation (2). For a long enough trajectory, finding the exact time-reverse is very unlikely (up to impossible in the extreme case of a trajectory that is as long as the observation time).

Although this challenge has been tackled before (see for example, E. Roldán, Irreversibility and dissipation in microscopic systems, Springer Theses, Berlin, 2014, S. Tusch, A. Kundu, G. Verley, T. Blondel, V. Miralles, D. Démoulin, D. Lacoste, and J. Baudry, Phys. Rev. Lett. 112, 180604, 2014, and D. Andrieux, P. Gaspard, S. Ciliberto, N. Garnier, S. Joubaud, and A. Petrosyan, J. Stat. Mech., P01002, 2008), it is not required for the calculation.

In order to calculate the individual terms in the sum that appears in equation 4, one needs to collect statistics on two consecutive jumps. Namely, if we see the sequence of transitions $\alpha \rightarrow \beta \rightarrow \mu$, we need to record the waiting times distribution for the transition $\beta \rightarrow \mu$, and for the transition $\beta \rightarrow \alpha$.

When applied to the hidden network case, this implies looking at $1 \rightarrow H \rightarrow 2$, and $2 \rightarrow H \rightarrow 1$, whereas when applied to the molecular motor case, this implies looking at two consecutive upwards or downward jumps.

To better clarify this point in the main text we have added the following sentence after equation 6:

“Let us emphasize that the calculation of \dot{S}_{WTD} requires collecting statistics on sequences of two consecutive jumps, i.e., $i \rightarrow j \rightarrow k$.”

Reviewer #3 (Remarks to the Author):

For general semi-Markov processes, the Authors are showing in this manuscript that the Kullback-Leibler divergence between the probabilities to observe a trajectory and the time-reversed trajectory has two non-negative contributions: (1) the well-known contribution from the transitions between the states of the stochastic process and (2) another contribution coming from the waiting time distributions of the process. Consistently, the semi-Markov process is reversible if and only if the two contributions are equal to zero. The contribution (2) is a new result of great importance for the study of irreversibility in many different contexts. In the absence of observable currents, the standard contribution (1) can be vanishing, so that the contribution (2) may still provide a lower bound on the thermodynamic entropy production, if this latter is related to the Kullback-Leibler divergence.

The manuscript is clearly written and address a broad audience. By the importance of the experimental relevance and the novelty of the result, this manuscript clearly deserves publication in Nature Communications.

Authors' response: We thank the reviewer for the support, and for time and effort invested in the review process. We appreciate the kind words, and the strong recommendation for publication.

There are a few typos to be fixed, e.g., "An observer unable resolve..." → "An observer unable to resolve..."; "In order to asses..." → "In order to assess..."; "less then..." → "less than..."; there is a weird symbol for the identity matrix in (B5) and (B7).

Authors' response: We thank the reviewer for pointing out those typos. They are now fixed in the revised version.

Reviewer #1 (Remarks to the Author):

The authors have clearly and satisfactorily addressed my main concerns. I now recommend publication of the manuscript in Nature comm.

Reviewer #2 (Remarks to the Author):

While I fully support the publication of the paper, one minor point is the following, Authors write in motivation: For instance, in the case of isothermal molecular motors hydrolyzing ATP ... What I believe is that in such active processes there is no usual meaning of the temperature T . In the presence of ATP or motors things become far from equilibrium. I might be wrong, so I leave this issue to the authors. But it seems that the authors claim that active transport is described close to thermal equilibrium. It is better if authors stick to math and not over sell.

Conclusion: accept the manuscript, no need for me to review the paper again.

Reviewer #3 (Remarks to the Author):

In my opinion, this manuscript should be published as such in Nature Communications.